# Nicotine and Its Downstream Metabolites in Maternal and Cord Sera: Biomarkers of Prenatal Smoking Exposure Associated with Offspring DNA Methylation

**DOI:** 10.3390/ijerph17249552

**Published:** 2020-12-20

**Authors:** Parnian Kheirkhah Rahimabad, Thilani M. Anthony, A. Daniel Jones, Shakiba Eslamimehr, Nandini Mukherjee, Susan Ewart, John W. Holloway, Hasan Arshad, Sarah Commodore, Wilfried Karmaus

**Affiliations:** 1Division of Epidemiology, Biostatistics, and Environmental Health, School of Public Health, University of Memphis, Memphis, TN 38152, USA; sslmmehr@memphis.edu (S.E.); nmkhrjee@memphis.edu (N.M.); karmaus1@memphis.edu (W.K.); 2Department of Biochemistry & Molecular Biology, Michigan State University, East Lansing, MI 48824, USA; thilani@chemistry.msu.edu (T.M.A.); jonesar4@msu.edu (A.D.J.); 3Department of Large Animal Clinical Sciences, Michigan State University, East Lansing, MI 48824, USA; ewart@cvm.msu.edu; 4Human Development and Health, Faculty of Medicine, University of Southampton, Southampton SO17 1BJ, UK; j.w.holloway@soton.ac.uk; 5Clinical and Experimental Sciences, Faculty of Medicine, University of Southampton, Southampton SO17 1BJ, UK; s.h.arshad@soton.ac.uk; 6The David Hide Asthma and Allergy Research Centre, Isle of Wight, Newport PO30 5TG, UK; 7NIHR Southampton Biomedical Research Centre, University Hospital Southampton, Hampshire SO16 6YD, UK; 8Department of Environmental and Occupational Health, Indiana University, Bloomington, IN 47405, USA; commodad@musc.edu

**Keywords:** prenatal smoking exposure, nicotine, maternal serum, cord serum, DNA methylation

## Abstract

Nicotine is a major constituent of cigarette smoke. Its primary metabolite in maternal and cord sera, cotinine, is considered a biomarker of prenatal smoking. Nicotine and cotinine half-lives are decreased in pregnancy due to their increased rate of metabolism and conversion to downstream metabolites such as norcotinine and 3-hydroxycotinine. Hence, downstream metabolites of nicotine may provide informative biomarkers of prenatal smoking. In this study of three generations (F0-mothers, F1-offspring who became mothers, and F2-offspring), we present a biochemical assessment of prenatal smoking exposure based on maternal and cord sera levels of nicotine, cotinine, norcotinine, and 3-hydroxycotinine. As potential markers of early effects of prenatal smoking, associations with differential DNA methylation (DNAm) in the F1- and F2-offspring were assessed. All metabolites in maternal and cord sera were associated with self-reported prenatal smoking, except for nicotine. We compared maternal self-report of smoking in pregnancy to biochemical evidence of prenatal smoking exposure. Self-report of F0-mothers of F1 in 1989–1990 had more accuracy identifying prenatal smoking related to maternal metabolites in maternal serum (sensitivity = 94.6%, specificity = 86.9%) compared to self-reports of F1-mothers of F2 (2010–2016) associated with cord serum markers (sensitivity = 66.7%, specificity = 78.8%). Nicotine levels in sera showed no significant association with any DNAm site previously linked to maternal smoking. Its downstream metabolites, however, were associated with DNAm sites located on the *MYO1G*, *AHRR*, and *GFI1* genes. In conclusion, cotinine, norcotinine, and 3-hydroxycotinine in maternal and cord sera provide informative biomarkers and should be considered when assessing prenatal smoking. The observed association of offspring DNAm with metabolites, except for nicotine, may imply that the toxic effects of prenatal nicotine exposure are exerted by downstream metabolites, rather than nicotine. If differential DNA methylation on the *MYO1G*, *AHRR*, and *GFI1* genes transmit adverse effects of prenatal nicotine exposure to the child, there is a need to investigate whether preventing changes in DNA methylation by reducing the metabolic rate of nicotine and conversion to harmful metabolites may protect exposed children.

## 1. Introduction

Maternal smoking in pregnancy is associated with adverse outcomes in offspring such as low birthweight, prematurity, and neonatal mortality [1,2,3]. It is conventionally assessed by maternal self-report of active and passive smoking during pregnancy. However, this approach could inaccurately estimate exposure because pregnant women may be reluctant to admit active smoking, have difficulty remembering smoking details, or are unaware of their secondhand smoke exposure [4].

Nicotine is the most studied toxic constituent of tobacco smoke with highly addictive properties [5,6]. Following cigarette smoke inhalation, nicotine is absorbed through small lung airways and alveoli and reaches a peak concentration in blood [5]. Nicotine is then metabolized by the liver. About 80% of nicotine is converted to cotinine, which is further converted to downstream metabolites such as norcotinine and 3-hydroxycotinine [5,7]. During pregnancy, nicotine from maternal circulation crosses the placenta and enters the fetal circulation. Maternal and cord sera nicotine and cotinine have been used in some studies to assess prenatal smoking exposure [7,8,9,10]. However, downstream metabolites of nicotine and cotinine in maternal and cord sera which may provide informative biomarkers due to increased rate of nicotine and cotinine metabolism in pregnancy [11] have not yet been measured and integrated into risk assessments of prenatal smoking exposure.

Nicotine, besides serving as a biomarker of smoking exposure, has been associated with many adverse effects of prenatal tobacco exposure such as impaired pulmonary, cardiovascular, and neurologic development [6,12,13]. Given the short half-life of nicotine in pregnancy in the mother due to its accelerated metabolism [14], it is possible that adverse outcomes linked to nicotine may be in fact a result of its downstream metabolites, which have longer half-lives [12] in maternal and fetal blood. On the other hand, it is possible that nicotine, cotinine, and their metabolites exert distinct effects on the fetus. In fact, studies on smokers have shown different and even antagonizing physiologic effects of nicotine and cotinine [15].

Recent studies on associations of prenatal smoking with offspring outcomes focus on epigenetic modifications such as DNA methylation (DNAm) as early markers and potential mediators between exposure and health outcomes. Multiple studies of the relationship of prenatal smoking and offspring DNAm were meta-analysed by Joubert et al., identifying 568 DNAm sites related to maternal gestational smoking self-report [16]. However, there is a lack of studies regarding the epigenetic effects of nicotine metabolites on offspring DNAm. Knowledge of the strength and directions of associations between newborn methylation levels at specific cytosine-phosphate-guanine sites (CpGs) and prenatal nicotine metabolites may provide potential information regarding downstream effects of each nicotine metabolite.

In this study, we aimed to address three issues regarding maternal smoking in pregnancy. Firstly, via a biochemical-based approach for evaluation of prenatal nicotine exposure using nicotine, cotinine, norcotinine, and 3-hydroxycotinine levels, we assessed the accuracy of maternal self-report of gestational smoking compared to chemical evidence of nicotine metabolites in maternal and cord sera. Secondly, we aimed to determine whether nicotine and its metabolites in maternal and cord sera can be used as biomarkers of smoking during gestation. Finally, we aimed to evaluate associations of maternal and cord sera nicotine, cotinine, norcotinine, and 3-hydroxycotinine with DNAm of the newborn (as early markers and potential mediators of health outcomes), focusing on 568 CpGs known to be associated with maternal smoking [16]. We used data from three generations of the Isle of Wight cohort: grandmothers (F0), mothers (F1), and F2-offspring [17,18].

## 2. Materials and Methods

### 2.1. Study Population

The IOW cohort was established in UK in 1989 to study asthma and allergic disorders [17]. The local research ethics committee (NRES Committee South Central—Hampshire B, UK) and University of Memphis Institutional Review Board in Memphis, U.S. (STUDY #: 2423) approved the IOW study. All participants at recruitment and follow-ups gave written consents. The IOW cohort includes three generations: F0-parents of the original cohort, F1-original cohort members, and F2, the offspring of F1. The current study focused on F1 generation and their F0 mothers (data collected in 1989–1990) and the F2 generation (data collected in 2010–2019) and their mothers (from F1 or female spouses of F1).

The F1 generation includes 1456 subjects from which 583 newborns remained for analysis after excluding those with missing values for maternal serum nicotine and its metabolites during pregnancy [17]. The F2 generation consists of 543 newborns from 331 mothers [18] of which 234 had biochemical data on nicotine metabolites in cord sera. Among these, data on DNAm at birth were available for 173 newborns.

Cigarette smoking information on mothers of the F1 and F2 generations was obtained using questionnaires during pregnancy or after giving birth. F0-mothers of the F1 generation were asked about active smoking during pregnancy. For F1-mothers of the F2 generation, more detailed information on active smoking during pregnancy was assessed by asking if they have smoked in the first (1–12 weeks), second (13–24 weeks), or third trimester (25 weeks to delivery) of pregnancy. We grouped active smoking mothers into persistent smokers (smoked throughout pregnancy), early-pregnancy smokers (smoked in trimesters one or two but stopped before trimester three), and inconsistent smokers (smoked in trimester three plus either trimester one or two). Passive smoking exposure during pregnancy was assessed by asking the mothers if they were regularly exposed to secondhand cigarette smoking while pregnant.

Maternal weight and height were measured in early pregnancy. Maternal age at delivery and newborns’ gender were obtained from hospital records. Socioeconomic status (SES) of the F0-generation was a derived variable using three indicators: the British socioeconomic classes based on parental occupation (1–6), family income, and number of children in the index child’s bedroom [19]. For the F1 generation, SES was defined based on household income, number of rooms in the house, and level of maternal education. Using cluster analysis, SES was categorized into five levels from lowest to highest. The highest, lowest, and three categories in the middle were taken as high, low, and intermediate SES, respectively.

Maternal blood samples from F0 participants were collected at birth. Sera from F0-mothers were aliquoted to measure nicotine and its metabolites. Heel prick blood samples of F1 on Guthrie cards were used to assess DNAm in F1 neonates at birth. For the F2-generation, cord blood samples were collected after delivery. Serum was aliquoted to measure nicotine and its metabolites; cell samples were collected to determine DNAm.

### 2.2. Sample Preparation and Processing

Serum specimens were grouped, processed, and analyzed in random order. Each batch included analyses of multiple blanks, pooled quality control extracts, and extracts of reference serum. Sera (20 µL aliquots) were extracted using a modified Matyash protocol [20] into water-soluble and organic-soluble fractions with each extraction tube containing 25 pmol cotinine-d3 as internal standard plus additional stable isotope-labeled internal standards (details in Appendix A). The polar (lower) fraction was evaporated to dryness under vacuum using a SpeedVac without heat application, and residues were dissolved in 200 µL of acetonitrile/water (9:1 *v/v*) and transferred to a glass auto-sampler vial with glass 200-µL insert.

### 2.3. Profiling of Nicotine and Its Metabolites Using Liquid Chromatography/High Resolution Mass Spectrometry

Profiling of polar fraction metabolites was executed using a QExactive mass spectrometer (Thermo Electron North America LLC, Madison, WI, USA) interfaced to a Thermo Vanquish Flex binary pump and auto-sampler equipped with an Acquity BEH Amide column (10 cm × 1.0 mm, 1.7 µm, Waters, Milford, MA, USA) for HILIC chromatographic separation, with analysis performed in positive-ion mode using full scan/all-ions fragmentation (additional details in Appendix A). Using all-ions fragmentation, chromatographic separations were performed at 30 °C.

### 2.4. Processing of LC/HRMS Data

Progenesis QI v2.4 software (Waters, Nonlinear Dynamics, Newcastle upon Tyne NE1 2JE, UK) was used for peak alignment, detection, normalization, and annotation. Annotations were suggested by searching spectra extracted using Compound Discoverer software (Thermo) against the mzCloud database (Thermo), followed by manual verification of characteristic fragment ions presence in the high collision-energy mass spectra. Peak areas were exported from Progenesis software and filtered to remove signals with highest abundances in blanks and those with relative mass defect (RMD) > 1200 ppm, as these are often ascribed to inorganic salts. Exported peak areas were normalized to the area of the internal standard cotinine-*d_3_* and scaled by multiplication by 1 × 104. The concentration of cotinine-*d_3_* added to each serum was 1.25 µM. Nicotine and its metabolites are assumed to have identical response factors owing to their similar physical and chemical properties, so the levels normalized signals of nicotine and its metabolites were multiplied by 0.125 to convert to nM concentrations in serum.

### 2.5. DNAm Measurement

For the F1-generation, DNA was isolated from dried blood spots obtained from heel prick tests on Guthrie cards at birth. QIAamp DNA Investigator kits (Qiagen Inc, Germantown, MD, USA) were used to extract DNA from 28mm^2^ samples punched from each Guthrie card per the manufacturer’s protocol, with incubation times at a minimum of 3 hr to overnight found to be optimal at 55 °C. DNA concentrations were measured using a Qubit spectrophotometer and high sensitivity standards and samples with concentrations ≥ 1.14 ng/ul were further processed for methylation. DNAm was measured using either Infinium MethylationEPIC BeadChip arrays or Infinium Human Methylation450K BeadChip arrays (Illumina Inc, San Diego, CA, USA). There were 8 total batches of F1-generation methylation data, two from the 450K array platform and six from the EPIC array platform. The CPACOR pipeline [21] was utilized for quality control (QC) and pre-processing of the quantile normalized beta values from the DNAm samples. Since different batches in EPIC arrays have varying feature numbers, there were pre-processed separately and combined with pre-processed samples from the 450K array. Only the shared probes between EPIC and 450K arrays were further analyzed. ComBat [22] was used to remove the batch effect in the combined dataset. In F2, DNA was extracted from cord blood samples by a standard salting out procedure [23]. One microgram of DNA was bisulfite-treated for cytosine to thymine conversion as stated by the manufacturer’s standard protocol using EZ 96-DNA methylation kit (Zymo Research, CA, USA) for each sample. DNAm was determined by the Illumina Infinium HumanMethylation450 Beadchip (Illumina, Inc., CA, USA). A standard protocol was used to process the arrays [24]. Samples were allocated randomly on microarrays to control for batch effects. Beadchips were scanned by BeadStation. Methylation levels in beta values were ascertained for each CpG site using the Methylation module of BeadStudio software. Beta values (β = $\frac{\mathrm{methylated}}{\mathrm{methylated}\;+\;\mathrm{unmethylated}\;+\;c}$) designate the proportions of methylated over the sum of methylated and unmethylated loci where c is a constant to prevent dividing by zero [25]. Since blood is composed of different cell populations, we adjusted for cell type proportions to remove the confounding effect of cell heterogeneity on DNAm data measured from blood samples [26,27]. The proportions of blood cell types were estimated using the function “estimateCellCount” in R-package “minfi” [28] modified from Houseman approach [29]. In particular, for DNAm data from Guthrie cards, the cell proportions (CD4 + T, CD8 + T, B-cells, monocytes, natural killer cells, neutrophils, and eosinophils) were estimated using the reference panel from Houseman et al. 2012 [29,30]. For DNAm data from cord blood, white blood cell counts were generated using the reference panel from Bakulski et al. 2016 [31]. Hence, in cord blood of F2 generation, the estimated cell types included CD4 + T, CD8 + T, B-cells, monocytes, natural killer cells, neutrophils, and nucleated red blood cells.

### 2.6. Statistical Analysis

Characteristics of the analyzed subjects were compared to those of the whole cohort using Chi-square tests and t tests. To distinguish metabolites values from “background noise”, we performed k-means cluster analysis to determine cut-off points (in R 4.0.0, R Foundation, Vienna, Austria) using the four nicotine metabolites (biomarkers: nicotine, cotinine, norcotinine, and 3-hydroxylcotinine). Two to five cluster solutions were tested and the 5-cluster-solution was retained as the optimal solution in both maternal or cord sera based on minimized intra-cluster variation and maximized inter-cluster variation (Appendix A). The clusters with the lowest levels of metabolites in maternal or cord sera were chosen as the non-exposed clusters. Then, we used the maximum levels of metabolites in the non-exposed cluster as the cut-off points of exposure to the four nicotine metabolites. Newborns in whom levels of nicotine or its metabolites in maternal (F0 mothers of F1) or cord (F2) sera were higher than the cutoffs were considered exposed to nicotine in pregnancy (biochemical evidence of prenatal nicotine exposure). In order to identify the sensitivity and specificity of maternal self-report of smoking during pregnancy, self-reported smoking was compared to the biochemically-defined prenatal nicotine exposure based on the four biomarkers measured in maternal (F0) and cord sera (F2). For assessing the sensitivity and specificity of maternal report of smoking exposure during pregnancy, biochemical evidence of maternal smoking exposure was considered as the gold standard. Sensitivity of maternal report of smoking was calculated as the proportion of those with positive biomarkers of maternal smoking who also reported gestational smoking. Specificity was calculated as proportion of those with negative biomarkers who reported no gestational smoking. For the F1 generation, linear regression analyses (in SAS 9.4) were used to model the concentration of each nicotine metabolite in maternal sera (F0 mothers) as a function of maternal active and passive smoking exposure in pregnancy. Unlike in the F1 generation, we had siblings in the F2 generation whose nicotine metabolites in cord sera could be correlated. To address this concern, we treated them as repeated measurements. To this end, we used linear mixed model analyses to adjust for repeated measurements of nicotine metabolites in cord sera of siblings. Both linear regression and linear mixed models were adjusted for maternal age at delivery, maternal BMI, and newborn’s gender. SES was initially included in the models; however, it was dropped from the analyses since it did not change the association of metabolites with DNAm by more than 10%. Same linear regression models (F1) and linear mixed models (F2) were used to compare nicotine and its metabolites in pregnancies with female and male offspring.

In the meta-analysis by Joubert et al., 6073 statistically significant CpGs with were reported in association with maternal smoking in pregnancy. Among them, 568 met the strict Bonferroni threshold for statistical significance. In our study, to evaluate the association of nicotine biomarkers related to DNAm, we chose the 568 CpGs for further testing since they had the most reliable evidence with regard to maternal smoking [16]. Linear regression models were used to assess the association of each nicotine metabolite with methylation of the candidate CpGs adjusted for peripheral blood cell types, maternal age at delivery, maternal BMI, and newborn gender. False discovery rate (FDR) was applied to adjust for multiple testing when evaluating the associations of CpG sites with nicotine metabolites [32]. A *p*-value of ≤ 0.05 was considered significant. We used the Venn diagram creator (bioinformatics.psb.ugent.be/webtools/Venn/) to depict CpGs in association with each nicotine metabolites.

## 3. Results

### 3.1. Study Characteristics

The analyzed sample of the F1-generation included 288 male and 295 female newborns (Table 1). Mothers of 122 F1-newborns reported active smoking in pregnancy (20.9%). Of 574 mothers with available data, 230 reported passive smoking in pregnancy (40.1%). There were no significant differences between the characteristics of the total F1-generation and the analyzed sample except for passive maternal smoking exposure which was lower in the analyzed sample (*p* value = 0.02).

The analyzed sample of the F2-generation consisted of 121 male and 113 female newborns. Of 218 mothers with available data, 150 reported no smoking during pregnancy (64.1%). Of mothers who smoked during pregnancy, 16, 21, and 31 of them were early, inconsistent, and persistent smokers, respectively. Out of 227 newborns with all available data, mothers of 71 newborns (31.3%) reported passive smoking exposure in pregnancy. Apart from SES (*p* value = 0.04), no other significant difference was observed between the total F2 generation and the analyzed sample.

The distribution of nicotine and its metabolites were highly right-skewed. Nicotine had a positive moderate correlation with cotinine (Rs = 0.6 and 0.7 in maternal and cord sera, respectively) and positive low correlations with norcotinine (Rs = 0.4 and 0.5) and 3-hydroxycotinine (Rs = 0.3 and 0.3) (Table 2). Cotinine was highly positively correlated with norcotinine (Rs = 0.8 and 0.8) and moderately positively correlated with 3-hydroxycotinine (Rs = 0.7 and 0.6).

### 3.2. Cut-Off Points of Nicotine and Related Metabolites in Maternal and Cord Sera

Cut-off points for prenatal exposure to cigarette smoking were based on the maximum values of nicotine and metabolites in a cluster with the lowest metabolites (Table 3). Compared to maternal serum metabolites of the F1-generation (from F0-mothers), self-reports of gestational smoking by mothers had a sensitivity of 94.6% and specificity of 86.9%, respectively. In the F2-generation, in comparison with cord serum metabolites, the sensitivity and specificity of maternal self-report of any gestational smoking were 66.7% and 78.8%, respectively. In the F2-generation, maternal self-report of smoking in the third trimester had a sensitivity of 62.5% and specificity of 87.1% (Table 4).

### 3.3. Nicotine and Related Metabolites in Maternal and Cord Sera as Biomarkers of Prenatal Smoking Exposure

In the F1-generation, nicotine levels in maternal serum (F0 mothers of F1) showed no significant association with maternal active smoking (Table 5). However, its metabolites including cotinine, norcotinine, and 3-hydroxycotinine were significantly associated with maternal active smoking in pregnancy, adjusted for newborn’s gender, maternal age, BMI, and self-report of passive smoking exposure in pregnancy (Table 5). In the F2-generation, cord serum nicotine also showed no significant association with active maternal smoking. Cotinine, norcotinine, and 3-hydroxycotinine measured in cord serum were significantly associated with persistent maternal smoking in pregnancy but not with smoking in early pregnancy or inconsistent smoking during pregnancy. None of the metabolites in maternal or cord sera were associated with passive maternal smoking exposure, after taking maternal active smoking exposure and confounding variables into account. Additionally, since there is suggestion of sexual dimorphisms of nicotine metabolism in the literature, we explored nicotine and other metabolites in male and female newborns. There was a non-significant trend of higher nicotine and lower downstream metabolites in male newborns compared to female newborns in F2-cord serum, but not in maternal serum of F0 mothers of F1. 

### 3.4. Nicotine and Related Metabolites in Maternal and Cord Sera and DNAm

Based on multiple studies, the meta-analysis from Joubert et al. identified 568 CpG sites that were significantly associated with maternal smoking (using maternal report or cotinine in maternal serum) after FDR adjustment [16]. These were candidates for testing associations with nicotine metabolites. In F1-newborns, methylation data of 460 of the 568 CpGs were available. Maternal serum nicotine showed no significant association with any CpGs. However, cotinine, norcotinine, and 3-hydroxycotinine in maternal serum were associated with the methylation levels of 31, 43, and 29 CpGs, respectively (Appendix A), adjusting for confounders, and correcting for FDR. Regarding genes on which the CpGs are located, maternal serum levels of cotinine, norcotinine, and 3-hydroxycotinine were associated with methylation of 19, 44, and 15 genes, respectively (Appendix A).

In F2-offspring, methylation data of 515 (of 568) CpGs were available. Again, cord serum nicotine levels showed no significant association with any of the CpGs. Adjusting for confounders and FDR, cord serum cotinine, norcotinine, and 3-hydroxycotinine had significant associations with 14, 12, and 7 CpGs, respectively. Among the CpGs that were significantly associated with cotinine, norcotinine, and 3-hydroxycotinie, seven, eight, and four of them were replicated in both F1- and F2-generations, respectively (Table 6). Cotinine, norcotinine, and 3-hydroxycotinine in cord sera were associated with methylation levels of six, five, and one genes, respectively in the newborns. The association of metabolites with CpGs in this study were all in the same direction as those in Joubert et al. meta-analysis [16]. There was substantial overlap between the CpGs associated with cotinine, norcotinine, and hydroxycotinine levels in maternal and cord sera (Appendix A).

## 4. Discussion

We assessed nicotine and its metabolites in maternal and cord sera corresponding with maternal smoking in pregnancy and the associated changes in DNAm of the offspring. First, we identified cut-off points for nicotine, cotinine, norcotinine, and 3-hydroxycotinine in maternal and cord sera indicating prenatal smoking exposure. These cut-off points separate biochemical evidences of nicotine exposure from background noise. Then, we compared biochemical evidence of prenatal smoking exposure (in maternal and cord sera) to maternal self-report of smoking in pregnancy (as gold standard). Maternal self-report had more accuracy identifying prenatal smoking exposure evidence in maternal serum (sensitivity = 94.6%, specificity = 86.9%) compared to cord serum (sensitivity = 66.7%, specificity = 78.8%). When assessing maternal and cord sera nicotine, cotinine, norcotinine, and 3-hydroxycotinine as biomarkers of prenatal smoking, all downstream nicotine metabolites, but not nicotine itself, were significantly associated with maternal active smoking self-reports. Lastly, we evaluated the associations of nicotine metabolites in maternal and cord sera with DNAm of the newborn as early markers and mediators of downstream health effects. Interestingly, nicotine showed no significant association with any of the CpGs that were previously linked to prenatal smoking exposure. However, nicotine metabolites (cotinine, norcotinine, and 3-hydroxycotinine) in maternal and cord sera were significantly associated with 103 and 33 CpGs, respectively (Appendix A). CpGs linked with three genes *MYO1G*, *AHRR*, and *GFI1* were associated with cotinine and norcotinine in maternal and cord sera. 3-hydroxycotinine was associated with CpGs linked to *GFI1*.

Previous studies have reported varying cut-offs of maternal serum cotinine indicating maternal smoking in pregnancy ranging from 17 nM to 99 nM [33]. Unlike previous studies [33,34,35,36], we used cut-off points of prenatal smoking exposure based on biochemical measurements rather than maternal self-report. The cut-off point of maternal cotinine levels in serum for identification of prenatal tobacco smoke exposure was 11 nM in our study compared to 30 nM in a previous study by Kvalvik et al. [33]. Since the rate of nicotine metabolism increases throughout pregnancy [37,38], different timing of maternal serum assessment (delivery in our study vs. 18th week of pregnancy) and reference used in setting the cut-off points (maternal self-report in Kvalvik et al.) may explain the variability between cut-off points in two studies. Future studies are necessary to further evaluate cut-off points of nicotine and its metabolites in maternal and cord sera based on chemical evidence of nicotine exposure in different months of pregnancy.

In our study, maternal self-report of smoking was related to a higher discrepancy with nicotine metabolites in cord serum than with those in maternal serum. This finding suggests that maternal self-report of smoking during pregnancy may not provide a good indicator of fetal exposure to nicotine metabolites in utero. Nicotine metabolites in cord serum may provide more sensitive indicators of prenatal nicotine exposure than nicotine metabolites in maternal serum, possibly due to longer half-life of nicotine in neonatal circulation [7]. Thus, the fetus could be exposed to nicotine metabolites longer than the mother. Relying solely on maternal self-reports in pregnancy, which was close to maternal serum nicotine metabolites levels in our study and previous reports [35], has been shown to underestimate negative effects of prenatal smoking exposure on the offspring [39].

Nicotine was moderately correlated with cotinine and lowly correlated with downstream metabolites, namely norcotinine and 3-hydroxycotinine (Table 2). On the other hand, the correlations between cotinine, norcotinine, and 3-hydroxycotinine were moderate to high suggesting the escalated conversion of nicotine to its downstream metabolites which makes them more accurate biomarkers of smoking in pregnancy.

In evaluating the association of prenatal nicotine exposure with DNAm, nicotine metabolites were measured in maternal and cord sera at delivery which is reflective of maternal exposure during late pregnancy. Throughout pregnancy, nicotine half-life is considered to decrease as its metabolic conversion rate to downstream metabolites increases [37,38]. The increased conversion rate of nicotine in late pregnancy may explain the observed association between nicotine metabolites, but not nicotine, and DNAm.

The substantial overlap of CpGs associated with cotinine, norcotinine, and hydroxycotinine levels in sera with the same direction of association implies that these metabolites have shared biological effects on the newborn. The observed association of cotinine and its downstream metabolites, but not nicotine, with smoking-related DNAm might be due to their better performances as biomarkers of prenatal smoking. Another explanation of the differing effects of nicotine compared to cotinine, norcotinine, and 3-hydroxycotinine on DNAm could be that methylation of affected genes results from the metabolic bystander products [40] during metabolism of nicotine or other constituents of tobacco smoke. Future research is warranted to investigate if the association of cotinine and downstream metabolites with offspring’s DNAm is a direct result of exposure to these metabolites or an indirect effect of smoking. Such research would have implications in addressing safety concerns regarding nicotine replacement therapy (NRT) in pregnant women.

Assuming that the effects of prenatal nicotine exposure are exerted by cotinine and downstream metabolites rather than nicotine, mediated by changes in DNAm, we expect to observe more detrimental effects of nicotine exposure in fast nicotine metabolizers. Inhibitors of nicotine metabolism have been developed as therapeutic means to decrease cigarette smoking dependency [41,42]. Animal studies may test whether inhibitors of nicotine metabolism alleviate the effects of prenatal nicotine exposure on the offspring.

Observed associations of downstream metabolites of nicotine with offspring DNAm may imply that the effects of prenatal nicotine exposure on offspring DNAm are exerted through changes in DNAm. The methylation of three genes affected by cotinine, norcotinine, and 3-hydroxycotinine in maternal and cord sera, namely *MYO1G*, *AHRR*, and *GFI1*, may have implications in downstream effects of prenatal nicotine exposure in the offspring.

The *MYO1G* gene codes for the plasma membrane-associated class I myosin mostly found in T- and B-lymphocytes and mast cells [43] controlling leukocyte mobility, adhesion, and phagocytosis [44]. Differential DNAm of CpGs linked to *MYO1G* gene may mediate the consequences of prenatal nicotine exposure such as increased leukocyte-endothelial adhesion (implicated in cardiovascular diseases) [44] and immunologic abnormalities [45]. The *AHRR* gene encodes for a protein involved in aryl hydrocarbon receptor signaling cascade participating in regulation of cell growth and differentiation and detoxifying xenobiotic substances [46]. *AHRR* has been implicated in lung function decline [47] and male infertility [48,49], two conditions associated with prenatal nicotine exposure [45,50].

The *GFI1* is another gene significantly associated with nicotine metabolites encoding a nuclear zinc finger protein involved in transcriptional repressing. Epigenetic alterations in *GFI1* gene have been found in development of cardiometabolic risks in offspring of smoker mothers [51]. The important role of these three genes is further suggested by replications of their associations with nicotine metabolites in two generations F1 and F2 (Table 6). Changes in DNAm at CpGs linked to *MYO1G*, *AHRR*, and *GFI1* genes may underlie the association of prenatal nicotine exposure to offspring including vascular, pulmonary, and cardiometabolic complications.

This study has some limitations. First, we did not have sufficient data on the number of cigarettes consumed per day or on any intervention received for quitting smoking such as NRT in smoking mothers. Second, the measured nicotine metabolites in maternal and cord sera were from different generations and collected at different years (1989–90 and 2010–2016). Future studies comparing these metabolites in the same mother-newborn dyads may provide better understanding of the differences between maternal and cord sera in terms of nicotine exposure assessment. Third, among the 568 CpGs in the meta-analysis by Joubert et al., we had DNAm data on 460 and 515 CpGs in F1 and F2 generations, respectively. Future research investigating a broader number of CpGs in larger samples may reveal more CpGs and genes associated with nicotine metabolites in maternal and cord sera. Fourth, we may discover additional CpGs and genes affected by maternal smoking if an epigenome-wide screening was conducted using biochemical markers. However, the current study was not designed for novel discoveries (due to sample size limitation) but relied on established markers. Finally, additional path-analytical approaches should be conducted to test links between prenatal nicotine exposure via biochemical markers and DNA methylation to health outcomes.

## 5. Conclusions

The findings of this study suggest that cotinine, norcotinine, and 3-hydroxycotinine in maternal and cord sera provide more accurate indicators of prenatal nicotine exposure than serum nicotine or maternal self-reports of smoking in pregnancy. Cotinine, norcotinine, and 3-hydroxycotinine in maternal and cord sera are associated with DNAm of the *MYO1G*, *AHRR*, and *GFI1* genes. These genes have been implicated in health conditions previously associated with prenatal nicotine exposure such as cardiovascular, pulmonary, and metabolic complications. Changes in DNAm may mediate the effects of prenatal nicotine exposure on later adverse health effects in offspring.

## Figures and Tables

**Table 1 ijerph-17-09552-t001:** Characteristics of study participants from the Isle of Wight cohort (F1 and F2 generations); comparison of total cohort with participants where samples were analyzed for nicotine metabolites in maternal (F0 mothers of F1) and cord (F2) sera.

**Characteristics**	**F1**		**F2**			
	Total cohort (n = 1456)	Analyzed sample (n = 583)	*p* value	Total cohort (n = 543)	Analyzed sample (n = 234)	*p* value
Newborn’s gender Female Male	(n = 1456) 50% 50%	(n = 583) 50.6% 49.4%	0.8	(n = 537) 44.4% 55.6%	(n = 234) 48.5% 51.5%	0.2
Maternal age at delivery (years)	(n = 1182) 29.6 (0.5)	(n = 504) 29.5 (0.4)	0.7	(n = 519) 24.3 (5.3)	(n = 233) 24.8 (4.6)	0.07
Maternal BMI (kg/m^2^)	(n = 1124) 23.5 (4.8)	(n = 468) 23.5 (4.6)	0.99	(n = 230) 25.6 (8.8)	(n = 230) 25.6 (8.8)	1
Active maternal smoking during pregnancy	(n = 1455) No: 75.5% Yes: 24.5%	(n = 583) No: 79.1% Yes: 20.9%	0.08	(n = 447) Non-smoker: 71.8% Early-pregnancy smoker:7.8% Inconsistent smoker: 7.2% Persistent smoker: 13.2%	(n = 218) Non-smoker: 68.8% Early-pregnancy smoker:7.4% Inconsistent smoker: 9.6% Persistent smoker: 14.2%	0.5
Maternal passive smoking during pregnancy	(n = 1430) 45%	(n = 574) 40.1%	0.02	(n = 400) 32.3%	(n = 227) 31.3%	0.7
Socioeconomic status Low Medium High	(n = 1340) 15.2% 76.6% 8.2%	(n = 567) 14.5% 76.7% 8.8%	0.8	(n = 398) 19.4% 68% 12.6%	(n = 190) 13.2% 76.3% 10.5%	0.04
Metabolite in maternal (F0 mothers of F1) and cord (F2) sera		(n = 583)			(n = 234)	
Nicotine (nM)	-	0.94 (2.32)	-	-	3.76 (7.49)	-
Cotinine (nM)	-	5.09 (9.17)	-	-	6.25 (10.57)	-
Norcotinine (nM)	-	0.04 (0.11)	-	-	0.11 (0.22)	-
3-Hydroxycotinine (nM)	-	0.19 (0.31)	-	-	0.10 (0.25)	-

Non-normally distributed variables (maternal age and BMI and metabolites) are reported as median (interquartile range). A *p* value < 0.05 shows a significant difference in characteristics between the total cohort and analyzed cohort.

**Table 2 ijerph-17-09552-t002:** Spearman correlation coefficients of nicotine and its metabolites (nM) in maternal (F0 mothers of F1 generation) and cord sera (F2 generation).

Metabolite	Maternal Serum (F0 Mothers of F1) Spearman Correlation Coefficients *			Cord Serum (F2) Spearman Correlation Coefficients *		
	Cotinine	Norcotinine	Hydroxycotinine	Cotinine	Norcotinine	Hydroxycotinine
Nicotine	0.7	0.5	0.3	0.6	0.4	0.3
Cotinine	-	0.8	0.6	-	0.8	0.7
Norcotinine	-	-	0.5	-	-	0.8

* All *p* values< 0.0001.

**Table 3 ijerph-17-09552-t003:** Cut-off points for nicotine and its metabolites in maternal (F0 mothers of F1 generation) and cord sera (F2 generation) based on the lowest cluster from cluster-analysis.

**Metabolite**	**Maternal Serum (F0 mothers of F1)**		
	**Median (IQR)**	**Cut-Off Points (nM)**	**Cut-Off Points (ng/mL)**
Nicotine	0.95 (2.3)	2.52	0.41
Cotinine	5.1 (24.15)	10.59	1.87
Norcotinine	0.04 (0.11)	0.14	0.022
Hydroxycotinine	0.19 (0.3)	0.54	0.103
**Metabolite**	**Cord Serum (F2)**		
	**Median (IQR)**	**Cut-Off Points (nM)**	**Cut-Off Points (ng/mL)**
Nicotine	3.75 (7.5)	0.77	0.12
Cotinine	6.25 (10.6)	1.29	0.23
Norcotinine	0.11 (0.22)	0.13	0.02
Hydroxycotinine	0.1 (0.25)	0.15	0.028

IQR: Interquartile range.

**Table 4 ijerph-17-09552-t004:** Sensitivity and specificity of maternal self-report of active smoking in pregnancy compared to biochemical evidence of nicotine exposure in maternal (F0 mothers of F1 generation) and cord (F2 generation) sera, respectively.

Maternal Self-Report	Sensitivity	Specificity
F1 generation (n = 583)		
Maternal active smoking	94.6%	86.9%
F2 generation (n = 218)		
Maternal active smoking	66.7%	78.8%
Late pregnancy (trimester 3) maternal active smoking	62.5%	87.1%

**Table 5 ijerph-17-09552-t005:** Association of nicotine and its metabolites in maternal (F0 mothers of F1) and cord (F2) sera with gestational active smoking and passive smoke exposure (coefficients derived from linear regressions) *.

Gestational Smoking Exposure	Nicotine (nM)		Cotinine (nM)		Norcotinine (nM)		3-hydroxycotinine (nM)	
	Estimate	*p* Value	Estimate	*p* Value	Estimate	*p* Value	Estimate	*p* Value
**F1 generation (n = 404)** **^ǂ^**								
Maternal active smoking	2.05	0.1	101.6	<0.0001	1.15	<0.0001	15.4	<0.0001
Maternal passive smoking exposure	0.35	0.7	1.4	0.8	0.015	0.9	0.04	0.97
**F2 generation (n = 152)** **^ǂ^**								
**Active maternal smoking ^#^**								
Early-pregnancy smoker	−2.3	0.3	14.8	0.1	0.1	0.6	1.85	0.4
Inconsistent smoker	0.65	0.7	13.3	0.1	0.2	0.1	3.2	0.07
Persistent smoker	−0.15	0.9	73.15	<0.0001	1.1	<0.0001	14.55	<0.0001
Maternal passive smoking exposure	2.02	0.2	1.9	0.7	0.05	0.5	0.04	0.98

* Linear regression models were adjusted for newborn’s gender, maternal age, and maternal BMI, and maternal self-report of active and passive smoking exposure in pregnancy (reference groups: non-smoker mothers and mothers who reported no secondhand smoking exposure in pregnancy); ^#^ Early-pregnancy smokers; mothers who smoked in trimesters one or two but stopped before trimester three. Inconsistent smokers; mothers who smoked in trimester three plus either trimester one or two. Persistent smokers: mothers who smoked throughout the three trimesters; ^ǂ^ Subjects with missing values for confounders were excluded from F1 (n = 583) and F2 (n = 234) generations.

**Table 6 ijerph-17-09552-t006:** CpGs and their linked genes linked statistically significantly associated with nicotine metabolites in maternal (F0 mothers of F1) and cord (F2) sera (discovered by Joubert et al. [16], tested in the F1 and replicated in the F2 generation).

Nicotine Metabolite	CpGs	Genes	Location on the Gene	Estimate (F1) *	FDR Adjusted *p* Value (F1)	Estimate (F2) *	FDR adjusted *p* Value (F2)
**Cotinine**	cg04180046	*MYO1G*	Body	0.0004	<0.0001	0.0006	0.05
	cg05575921	*AHRR*	Body	−0.0002	<0.0001	−0.0008	0.02
	cg09935388	*GFI1*	Body	−0.0004	0.002	−0.0012	0.02
	cg12803068	*MYO1G*	Body	0.0004	<0.0001	0.0008	0.05
	cg12876356	*GFI1*	Body	−0.0001	0.01	−0.0012	0.02
	cg14179389	*GFI1*	Body	−0.0004	0.001	−0.0006	0.05
	cg19089201	*MYO1G*	3′UTR	0.00018	0.0006	0.0004	0.05
**Norcotinine**	cg04180046	*MYO1G*	Body	0.04	<0.0001	0.04	0.05
	cg04535902	*GFI1*	Body	−0.04	0.01	−0.04	0.03
	cg05575921	*AHRR*	Body	−0.04	<0.0001	−0.04	0.01
	cg09662411	*GFI1*	Body	−0.04	0.02	−0.06	0.01
	cg09935388	*GFI1*	Body	−0.04	<0.0001	−0.06	0.01
	cg12876356	*GFI1*	Body	−0.06	0.0008	−0.08	0.01
	cg18146737	*GFI1*	Body	−0.04	0.02	−0.08	0.01
	cg18316974	*GFI1*	Body	−0.02	0.04	−0.06	0.03
**3-Hydroxycotinine**	cg04535902	*GFI1*	Body	−0.002	0.05	−0.004	0.03
	cg09935388	*GFI1*	Body	−0.004	0.0005	−0.004	0.04
	cg12876356	*GFI1*	Body	−0.004	0.007	−0.004	0.03
	cg18146737	*GFI1*	Body	−0.002	0.05	−0.006	0.03

* Estimates are beta values of DNA methylation.

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
