# Peer review of "Nicotine and Its Downstream Metabolites in Maternal and Cord Sera: Biomarkers of Prenatal Smoking Exposure Associated with Offspring DNA Methylation"

_ijerph, 2020, doi:10.3390/ijerph17249552_

Round 1
Reviewer 1 Report
In this study, the author aimed to address three issues regarding maternal smoking in pregnancy. Firstly, the author detected nicotine, cotinine, norcotinine, and 3-hydroxycotinine levels in maternal and cord serum, and assessed the accuracy of maternal self-report of gestational smoking compared to chemical evidence of nicotine metabolites in maternal and cord serum. Secondly, the author clarified whether nicotine and its metabolites in maternal and cord serum can be used as biomarkers of smoking during gestation. Finally, the author evaluated the associations of maternal and cord serum nicotine, cotinine, norcotinine, and 3-hydroxycotinine levels with DNAm of the newborn (including F1 and F2). The findings of this study suggested that cotinine, norcotinine, and 3-hydroxycotinine in maternal and cord serum provided more accurate indicators of prenatal nicotine exposure than serum nicotine or maternal self-reports of smoking in pregnancy. Cotinine, norcotinine, and 3-hydroxycotinine in maternal and cord serum were associated with DNAm of the genes (including MYO1G, AHRR, and GFI1). The DNAm of these genes may mediate the effects of prenatal nicotine exposure on later adverse health effects in offspring. This is a very interesting research. However, the following problems need to be clarified and modified. (1) In line 223, the author detected 568 CpGs associated with maternal smoking in pregnancy according to literature. However, the literature reported 6000 CpGs associated with maternal smoking in pregnancy. Why only 568 of them were detected? How many genes did the 568 CpGs correspond to? Please explain and add in it. (2) This study only mentioned nicotine and its three metabolites. In line 150, why the title is “Profiling of Metabolites, Nutrients, and Toxins using Liquid Chromatography/High Resolution Mass Spectrometry ”. A clerical error? (3) Which method was used to calculate the specificity and sensitivity was not explained in this study. Please add in it. (4) In DNA methylation detection (2.5), why detection method of F1 and F2 cord blood gene methylation were inconsistent? (5) Non-standard text such as 28mm2 in 173 lines? estimateCellCounts() in 185 lines? (6) The format of the whole references was not consistent, and the number of references was also not standard, please modify according to the requirements of the magazine.Author Response
We would like to appreciate the valuable comments and suggestions of the reviewers, which have considerably improved the manuscript. We provide our responses to the comments below:
Review 1:
In this study, the author aimed to address three issues regarding maternal smoking in pregnancy. Firstly, the author detected nicotine, cotinine, norcotinine, and 3-hydroxycotinine levels in maternal and cord serum, and assessed the accuracy of maternal self-report of gestational smoking compared to chemical evidence of nicotine metabolites in maternal and cord serum. Secondly, the author clarified whether nicotine and its metabolites in maternal and cord serum can be used as biomarkers of smoking during gestation. Finally, the author evaluated the associations of maternal and cord serum nicotine, cotinine, norcotinine, and 3-hydroxycotinine levels with DNAm of the newborn (including F1 and F2). The findings of this study suggested that cotinine, norcotinine, and 3-hydroxycotinine in maternal and cord serum provided more accurate indicators of prenatal nicotine exposure than serum nicotine or maternal self-reports of smoking in pregnancy. Cotinine, norcotinine, and 3-hydroxycotinine in maternal and cord serum were associated with DNAm of the genes (including MYO1G, AHRR, and GFI1). The DNAm of these genes may mediate the effects of prenatal nicotine exposure on later adverse health effects in offspring. This is a very interesting research. However, the following problems need to be clarified and modified.
(1) In line 223, the author detected 568 CpGs associated with maternal smoking in pregnancy according to literature. However, the literature reported 6000 CpGs associated with maternal smoking in pregnancy. Why only 568 of them were detected? How many genes did the 568 CpGs correspond to? Please explain and add in it.
The 568 CpGs corresponded to 368 genes. We added the following to the line 241 as a rationale for choosing the 568 CpGs: “In the meta-analysis by Joubert et al., 6,073 statistically significant CpGs with were reported in association with maternal smoking in pregnancy. Among them, 568 met the strict Bonferroni threshold for statistical significance. In our study, to evaluate associations of nicotine biomarkers related to DNAm, we chose the 568 CpGs for further testing since they had the most reliable evidence with regard to maternal smoking”
(2) This study only mentioned nicotine and its three metabolites. In line 150, why the title is “Profiling of Metabolites, Nutrients, and Toxins using Liquid Chromatography/High Resolution Mass Spectrometry ”. A clerical error?
This work was part of an investigation of the association of metabolites, nutrients, and toxins in maternal and cord sera with offspring DNA methylation and health outcomes. Nicotine and its three metabolites were part of these chemicals that we detected in sera. However, since our study focuses on nicotine and its metabolites, we changed the “Metabolites, Nutrients, and Toxins “ in title to “nicotine and its metabolites”.(line 151)
(3) Which method was used to calculate the specificity and sensitivity was not explained in this study. Please add in it.
The following was added to the methods (line 224):
“For assessing the sensitivity and specificity of maternal report of smoking exposure during pregnancy, biochemical evidence of maternal smoking exposure was considered as the gold standard. Sensitivity of maternal report of smoking was calculated as the proportion of those with positive biomarkers of maternal smoking who also reported gestational smoking. Specificity was calculated as proportion of those with negative biomarkers who reported not gestational smoking.”
(4) In DNA methylation detection (2.5), why detection method of F1 and F2 cord blood gene methylation were inconsistent?
In F1, DNA was extracted from Guthrie cards obtained from heel prick tests. In F2 generation, DNA was extracted from cord blood. Although both cord blood and heel prick tests reveal DNA methylation at birth, slightly different procedures were used to analyze DNA methylation from these sources and different cell-type corrections were applied since cord blood still include nucleated red blood cells.
In order to address the differences better, we removed the “We estimated cell types using the “estimateCellCounts”() function in Minfi package [26] and the Houseman et al. 2016 reference panel.” from line 186 and 187 and “The proportions of cord blood cell-types (nRBC, neutrophil, eosinophil, natural killer cell, CD4+ T cell, B cell) were estimated using Bioconductor and “minfi” R package based on reference values of cell-type-specific CpG sites.” From line 206.
Instead, we added the following to the line 196:
“Since blood is composed of different cell populations, we adjusted for cell type proportions to remove the confounding effect of cell heterogeneity on DNAm data measured from blood samples [28,29]. The proportions of blood cell types were estimated using the function “estimateCellCount” in R-package “minfi” [30] modified from Houseman approach [31]. In particular, for DNAm data from Guthrie cards, the cell proportions (CD4+T, CD8+T, B-cells, monocytes, natural killer cells, neutrophils, and eosinophils) were estimated using the reference panel from Houseman et al. 2012 [31,32]. For DNAm data from cord blood, white blood cell counts were generated using the reference panel from Bakulski et al. 2016 [33]. Hence, in cord blood of F2 generation, the estimated cell types included CD4+T, CD8+T, B-cells, monocytes, natural killer cells, neutrophils, and nucleated red blood cells.”
(5) Non-standard text such as 28mm2 in 173 lines? estimateCellCounts() in 185 lines?
In line 173 (now line 174), 28mm2 was changed to “28 mm2”. In line 185 (now line 199), “estimateCellCounts()” is a function in minfi R package. To resolve the confusion, we deleted the parenthesis and changed it to “estimateCellCounts” function.
(6) The format of the whole references was not consistent, and the number of references was also not standard, please modify according to the requirements of the magazine.
Thanks for detecting this. The whole references were modified and updated according to the MDPI style and journal’s requirements.

Reviewer 2 Report
Comments for Internal Journal of Environmental Research and Public Health manuscript titled “Nicotine and its downstream metabolites in maternal and cord sera: Biomarkers of prenatal smoking exposure associated with offspring DNA methylation”
General comments:
This paper focuses on establishing whether nicotine metabolites can be effectively used as biomarkers of smoking during pregnancy and subsequently fetal exposure to nicotine and examines the relationship between the presence of nicotine’s metabolites in both maternal and cord sear and DNA methylation. In general, the study is well written and clear however the presentation of the results specifically pertaining to the study population could be improved. The discussion is relatively well rounded but could include a bit more information in certain areas. This study adds further evidence to support that self-reporting of smoking during pregnancy is not a good indicator of smoking during pregnancy and provides evidence to support using cord blood as an effective method, more so that maternal serum, for determining fetal exposure to smoking in utero. That the presence of nicotine’s metabolites in cord blood are closely correlated to DNA methylation can help provide new understanding on nicotine’s effects on the fetus.
Minor comments
Introduction
- Line 82, the sentence should include an “are” between “that” and “capable” so that it reads “of chemicals that are capable of exerting”.
Materials and Methods
- When the smoking information was acquired from the participants, when they smoked and whether they were exposed to passive smoke was asked but was there any information at all about whether any of them used nicotine patches or gum as a replacement for smoking or as an avenue to eventually quit at some stage during the pregnancy? Was the amount of smoking/nicotine exposure quantified at all? There is a positive relationship between the magnitude of DNA methylation and the level of fetal exposure to cigarette smoke (Wiklund et al, 2019: DNA methylation links prenatal smoking exposure to later life health outcomes in offspring. Clinical Epigenetics 11:97).
Results:
- The way Table 1 is set out is very confusing, it took me a long time to process it and it simply came down to clarity. I would suggest making the columns for F1 and F2 more distinct, perhaps by widening the space between the two and dispensing with the N columns and just including the N value above the values for each characteristic. This lessens the number of columns in the Table and in my opinion will make the table easier to read. Please find an example below. Perhaps also state in the legend beneath the table that p<0.05 shows a significant difference in characteristics between the total cohort and analyzed cohort.
|
|
F1 |
||
|
Characteristics |
Total Cohort (n= 1456) |
Analyzed sample (n=583) |
P value |
|
Newborns gender Male Female |
n =1456
50% 50% |
n=583
50.6% 49.4% |
0.8
|
|
Maternal Age at delivery |
n= 1182 29.6 (0.5) |
n=504 29.5(0.4) |
0.7 |
- Should the value N value for characteristic “Newborns gender” under F2 be 543 instead of 537? Were some newborn data discarded from the total? If so I think this should be reflected in the text somewhere probably in paragraph 2 of the methods section 2.1: Study population. The way this N value is placed in the Table makes it look like you are indicating that 537 newborns were female, but this doesn’t fit with the math. I am having the same problem with the N values for Total and Analyzed cohorts for the same characteristic, hence why I suggest reorganising the Table.
- My understanding is that maternal sera was collected for F1 generation (that is the mothers of F1 generation, F0), but not F2 generation (the mothers of F2, F1) as indicated in the 2nd paragraph of the methods section 2.1: Study population. As such I am not sure why in the 2nd part of Table 1 under the characteristic for Metabolite in maternal and cord sera, you have written F0-F1 for maternal sera. Perhaps I have misunderstood where your samples have come from? In table 2 however you only refer to F0 maternal cord serum? Either I have misunderstood or there is some inconsistency between Tables. Is there a reason why maternal sera from F2 generation (the mothers of F2, F1) was not available/collected? It would have been interesting if you had been able to collect maternal sera from F2 (mothers pf F2, F1) and compare that with the cord sera obtained from the F2 generation.
- I am glad that you included results for a test comparing differences in how nicotine is metabolised between males and females. I do think you should mention that this test was conducted somewhere in the methods.
- Perhaps it might be useful to add a figure of your k-means cluster analysis and the clusters derived from that. Some additional info in text about how the data was prepared for the analysis (i.e did it need to be standardized?) would also be useful.
Discussion
- Is there currently a standardised cut-off point for distinguishing between smokers and non-smokers in pregnant women? Does your data add any strength to what this cut-off should be or is it customary for each study to find and use its own?
- Is there any current literature available about when during pregnancy the clearance rates and half-life of nicotine and its metabolites are increased and reduced respectively? Adding this information may add to your discussion regarding why cut-offs may differ when metabolite readings are taken at different stages of pregnancy. It may also explain why there is a relationship between smoking at a later stage of pregnancy (rather than at an earlier stage of pregnancy) and DNA methylation.
- Line 321: There seems to be a typographical error in this line. Delete “to” at the end of the line.

Author Response
We would like to appreciate the valuable comments and suggestions of the reviewers, which have considerably improved the manuscript. We provide our responses to the comments below:
Review 2:
General comments:
This paper focuses on establishing whether nicotine metabolites can be effectively used as biomarkers of smoking during pregnancy and subsequently fetal exposure to nicotine and examines the relationship between the presence of nicotine’s metabolites in both maternal and cord sear and DNA methylation. In general, the study is well written and clear however the presentation of the results specifically pertaining to the study population could be improved. The discussion is relatively well rounded but could include a bit more information in certain areas. This study adds further evidence to support that self-reporting of smoking during pregnancy is not a good indicator of smoking during pregnancy and provides evidence to support using cord blood as an effective method, more so that maternal serum, for determining fetal exposure to smoking in utero. That the presence of nicotine’s metabolites in cord blood are closely correlated to DNA methylation can help provide new understanding on nicotine’s effects on the fetus.
Minor comments
Introduction
- Line 82, the sentence should include an “are” between “that” and “capable” so that it reads “of chemicals that are capable of exerting”.
Thanks for the correction. The “are” is added.
Materials and Methods
- When the smoking information was acquired from the participants, when they smoked and whether they were exposed to passive smoke was asked but was there any information at all about whether any of them used nicotine patches or gum as a replacement for smoking or as an avenue to eventually quit at some stage during the pregnancy?
Unfortunately, no such question was asked from the mothers. However, we evaluated nicotine exposure in pregnancy using biochemical assessment of nicotine and its metabolites in sera which encompasses nicotine exposure from all sources. Following your comment, we added this as a limitation of our study (line 431):
“First, we did not have sufficient data on the number of cigarettes consumed per day or on any intervention received for quitting smoking such as NRT in smoking mothers.”
(NRT was defined as nicotine replacement therapy in previous lines in the manuscript.)
- Was the amount of smoking/nicotine exposure quantified at all? There is a positive relationship between the magnitude of DNA methylation and the level of fetal exposure to cigarette smoke (Wiklund et al, 2019: DNA methylation links prenatal smoking exposure to later life health outcomes in offspring. Clinical Epigenetics 11:97).
Some mothers reported the number of cigarettes per day, however, the number of mothers with detailed reports was not sufficient for the analysis since it widely varied during pregnancy. However, in evaluation of the association of maternal smoking in pregnancy and offspring DNA methylation, we focused on nicotine metabolites in sera rather than the maternal reports. However, nicotine metabolites levels in the serum are influenced by the amount of maternal smoking. Following the comment, we added this as a limitation to our study (line 431)
“First, we did not have sufficient data on the number of cigarettes consumed per day or on any intervention received for quitting smoking such as NRT in smoking mothers.”
Results:
- The way Table 1 is set out is very confusing, it took me a long time to process it and it simply came down to clarity. I would suggest making the columns for F1 and F2 more distinct, perhaps by widening the space between the two and dispensing with the N columns and just including the N value above the values for each characteristic. This lessens the number of columns in the Table and in my opinion will make the table easier to read. Please find an example below. Perhaps also state in the legend beneath the table that p<0.05 shows a significant difference in characteristics between the total cohort and analyzed cohort.
|
|
F1 |
||
|
Characteristics |
Total Cohort (n= 1456) |
Analyzed sample (n=583) |
P value |
|
Newborns gender Male Female |
n =1456
50% 50% |
n=583
50.6% 49.4% |
0.8
|
|
Maternal Age at delivery |
n= 1182 29.6 (0.5) |
n=504 29.5(0.4) |
0.7 |
Thank you so much for the suggestion. Table one was rearranged as the following (line 270):
Table 1. Characteristics of study participants from the Isle of Wight cohort (F1 and F2 generations); comparison of total cohort with participants where samples were analyzed for nicotine metabolites in maternal (F0-F1) and cord (F2) sera.
|
Characteristics |
|
F1 |
|
F2 |
|
||||||
|
Total cohort (n=1456) |
Analyzed sample (n=583) |
P value |
Total cohort (n=543) |
Analyzed sample |
P value |
|
|||||
|
Newborn’s gender Female Male |
|
(n=1456) 50% 50% |
(n=583) 50.6% 49.4% |
0.8 |
(n=537) 44.4% 55.6% |
(n=234) 48.5% 51.5% |
0.2 |
|
|||
|
Maternal age at delivery (years) |
|
(n=1182) 29.6 (0.5) |
(n=504) 29.5 (0.4) |
0.7 |
(n=519) 24.3 (5.3) |
(n=233) 24.8 (4.6) |
0.07 |
|
|||
|
Maternal BMI (kg/m2) |
|
(n=1124) 23.5 (4.8) |
(n=468) 23.5 (4.6) |
0.99 |
(n=230) 25.6 (8.8) |
(n=230) 25.6 (8.8) |
1 |
|
|||
|
Active maternal smoking during pregnancy
|
|
(n=1455) No: 75.5% Yes: 24.5%
|
(n=583) No: 79.1% Yes: 20.9%
|
0.08 |
(n=447) Non-smoker: 71.8% Early-pregnancy smoker:7.8% Inconsistent smoker: 7.2% Persistent smoker: 13.2% |
(n=218) Non-smoker: 68.8% Early-pregnancy smoker:7.4% Inconsistent smoker: 9.6% Persistent smoker: 14.2% |
0.5 |
|
|||
|
Maternal passive smoking during pregnancy |
|
(n=1430) 45% |
(n=574) 40.1% |
0.02 |
(n=400) 32.3% |
(n=227) 31.3% |
0.7 |
|
|||
|
Socioeconomic status Low Medium High
|
|
(n=1340) 15.2% 76.6% 8.2%
|
(n=567) 14.5% 76.7% 8.8% |
0.8
|
(n=398) 19.4% 68% 12.6% |
(n=190) 13.2% 76.3% 10.5% |
0.04 |
||||
|
Metabolite in maternal (F0 mothers of F1) and cord (F2) sera |
|
|
(n=583) |
|
|
(n=234) |
|
||||
|
Nicotine (nM) |
|
- |
0.94 (2.32) |
- |
- |
3.76 (7.49) |
- |
||||
|
Cotinine (nM) |
|
- |
5.09 (9.17) |
- |
- |
6.25 (10.57) |
- |
||||
|
Norcotinine (nM) |
|
- |
0.04 (0.11) |
- |
- |
0.11 (0.22) |
- |
||||
|
3-Hydroxycotinine (nM) |
|
- |
0.19 (0.31) |
- |
- |
0.10 (0.25) |
- |
||||
Non-normally distributed variables (maternal age and BMI and metabolites) are reported as median (interquartile range). A p value <0.05 shows a significant difference in characteristics between the total cohort and analyzed cohort.
- Should the value N value for characteristic “Newborns gender” under F2 be 543 instead of 537? Were some newborn data discarded from the total? If so I think this should be reflected in the text somewhere probably in paragraph 2 of the methods section 2.1: Study population. The way this N value is placed in the Table makes it look like you are indicating that 537 newborns were female, but this doesn’t fit with the math. I am having the same problem with the N values for Total and Analyzed cohorts for the same characteristic, hence why I suggest reorganising the Table.
Thanks for the correction. Table 1 is reorganized as suggested and 537 was corrected to 543.
- My understanding is that maternal sera was collected for F1 generation (that is the mothers of F1 generation, F0), but not F2 generation (the mothers of F2, F1) as indicated in the 2nd paragraph of the methods section 2.1: Study population. As such I am not sure why in the 2nd part of Table 1 under the characteristic for Metabolite in maternal and cord sera, you have written F0-F1 for maternal sera. Perhaps I have misunderstood where your samples have come from?
Yes. This is correct. F0-maternal sera were collected for F1 generation (that is the mothers of F1, namely F0). To solve the confusion, we changed the “F0-F1” in the table 1 to “F0 mothers of F1”.
- In table 2 however you only refer to F0 maternal cord serum? Either I have misunderstood or there is some inconsistency between Tables. Is there a reason why maternal sera from F2 generation (the mothers of F2, F1) was not available/collected? It would have been interesting if you had been able to collect maternal sera from F2 (mothers pf F2, F1) and compare that with the cord sera obtained from the F2 generation.
“F0-F1” in the table 1 was changed to “F0 mothers of F1” to be consistent with table 2 and other tables. Unfortunately, maternal serum samples from F1 mothers were not collected for assessment of nicotine and its metabolites.
- I am glad that you included results for a test comparing differences in how nicotine is metabolised between males and females. I do think you should mention that this test was conducted somewhere in the methods.
We added the following to the line 238:
“Same linear regression models (F1) and linear mixed models (F2) were used to compare nicotine and its metabolites in pregnancies with female and male offspring.”
- Perhaps it might be useful to add a figure of your k-means cluster analysis and the clusters derived from that. Some additional info in text about how the data was prepared for the analysis (i.e did it need to be standardized?) would also be useful.
Thank you for the suggestion. The figure was added as a supplementary figure (line 217). We addressed the process of standardization in the methods.
Discussion
- Is there currently a standardised cut-off point for distinguishing between smokers and non-smokers in pregnant women? Does your data add any strength to what this cut-off should be or is it customary for each study to find and use its own?
There is no standardized cut-off point for distinguishing between smokers and non-smokers in pregnant women. Previous studies have reported different cut-offs of maternal serum cotinine and maternal smoking in pregnancy from 17 nM to 99 nM. (reference: https://www.nature.com/articles/pr201236) Our study provides cut-off points for nicotine and its three metabolites both in maternal and in cord sera. In addition, we defined the cut-off based on chemical evidence of nicotine exposure rather than maternal reports.
Following your comment, we added the following to the discussion section where we discuss cut-off points for nicotine and its metabolites:
“Previous studies have reported varying cut-offs of maternal serum cotinine indicating maternal smoking in pregnancy ranging from 17 nM to 99 nM [33]”. (line 362)
And
“Future studies are necessary to further evaluate cut-off points of nicotine and its metabolites in maternal and cord sera based on chemical evidence of nicotine exposure in different months of pregnancy.” (line 370)
- Is there any current literature available about when during pregnancy the clearance rates and half-life of nicotine and its metabolites are increased and reduced respectively? Adding this information may add to your discussion regarding why cut-offs may differ when metabolite readings are taken at different stages of pregnancy. It may also explain why there is a relationship between smoking at a later stage of pregnancy (rather than at an earlier stage of pregnancy) and DNA methylation.
Thanks for the suggestion. Yes, there are some papers indicating an increased rate of nicotine metabolism throughout pregnancy. According to the suggestion, we add more details and information to the part of discussion on the cut-off points (line 362):
“Previous studies have reported varying cut-offs of maternal serum cotinine indicating maternal smoking in pregnancy ranging from 17 nM to 99 nM [33]. Unlike previous studies [33-36], we used cut-off points of prenatal smoking exposure based on biochemical measurements rather than maternal self-report. The cut-off point of maternal cotinine levels in serum for identification of prenatal tobacco smoke exposure was 11 nM in our study compared to 30 nM in a previous study by Kvalvik et al. [37]. Since the rate of nicotine metabolism increases throughout pregnancy [38,39], different timing of maternal serum assessment (delivery in our study vs. 18th week of pregnancy) and reference used in setting the cut-off points (maternal self-report in Kvalvik et al.) may explain the variability between cut-off points in two studies. Future studies are necessary to further evaluate cut-off points of nicotine and its metabolites in maternal and cord sera based on chemical evidence of nicotine exposure in different months of pregnancy.”
And the following to the line 387:
“In evaluating the association of prenatal nicotine exposure with DNAm, nicotine metabolites were measured in maternal and cord sera at delivery which is reflective of maternal exposure during late pregnancy. Throughout pregnancy, nicotine half-life is considered to decrease as its metabolic conversion rate to downstream metabolites increases [40,41]. The increased conversion rate of nicotine in late pregnancy may explain the observed association between nicotine metabolites, but not nicotine, and DNAm.”
- Line 321: There seems to be a typographical error in this line. Delete “to” at the end of the line.
Thank you for your comment. It was corrected.

Reviewer 3 Report
This is an interesting paper that presents a biochemical assessment of prenatal smoking exposure based on maternal and cord sera levels of nicotine, cotinine, norcotinine, and 3-hydroxycotinine. The authors suggest that the cotinine, norcotinine, and 3-hydroxycotinine in maternal and cord sera provide a more accurate indicator of prenatal nicotine exposure than serum nicotine or maternal-reports of smoking in pregnancy. Also, cotinine, norcotinine, and 3-hydroxycotinine in maternal and cord sera were associated with DNA methylation of the MYO1G, AHRR, and GFI1 genes. Some minor editorial input and clarifications are required:
- Line 82 is missing 'are' before capable. Also, it is not clear as to what is implied in this statement as downstream nicotine metabolites are the focus of this study in terms of how they may influence functional outcomes in the offspring through modulation of DNA methylation.
- Line 185 needs correction as the parenthesis is missing information.
- Line 321 has a 'to' at the end that seems misplaced.
- Line 331 seems to be missing the number of CpGs for one of the metabolites
- Line 373-74 is redundant as it says that "effects of prenatal nicotine exposure on offspring DNAm are exerted through changes in DNAm".
- In line 377-91, is there literature that shows how methylation changes on the reported genes in utero influence the long term health outcomes of the offspring? As is, not much is said about MYO1G with regards to fetal development and the long term health outcomes. This does not need to be based on maternal smoking.
- Also, since the role of methylation on gene transcription regulation is dependent on whether it occurs in the promoter region or gene body, has there been data indicating how smoking affect the direction of gene expression, especially with regard to these 3 genes?
- Also, since lifestyle and environmental factors are known to influence DNA methylation, especially during the fetal development stage, were factors such as nutrition and medications accounted for in the design?
Author Response
We would like to appreciate the valuable comments and suggestions of the reviewers, which have considerably improved the manuscript. We provide our responses to the comments below:
Review 3:
This is an interesting paper that presents a biochemical assessment of prenatal smoking exposure based on maternal and cord sera levels of nicotine, cotinine, norcotinine, and 3-hydroxycotinine. The authors suggest that the cotinine, norcotinine, and 3-hydroxycotinine in maternal and cord sera provide a more accurate indicator of prenatal nicotine exposure than serum nicotine or maternal-reports of smoking in pregnancy. Also, cotinine, norcotinine, and 3-hydroxycotinine in maternal and cord sera were associated with DNA methylation of the MYO1G, AHRR, and GFI1 genes. Some minor editorial input and clarifications are required:
Line 82 is missing 'are' before capable. Also, it is not clear as to what is implied in this statement as downstream nicotine metabolites are the focus of this study in terms of how they may influence functional outcomes in the offspring through modulation of DNA methylation.
Thanks for the correction. Missing “are” was added. However, since the information stated in the paragraph seems unnecessary and may confuse the readers, we deleted the paragraph from introduction (from line 78 to 84)
Line 185 needs correction as the parenthesis is missing information.
estimateCellCounts() is a function of minfi R package. To resolve the confusion, we deleted the parenthesis and changed it to “estimateCellCounts function”.
Line 321 has a 'to' at the end that seems misplaced.
Thanks for the comment. It was corrected.
Line 331 seems to be missing the number of CpGs for one of the metabolites
The original sentence says:” However, cotinine, norcotinine, and 3-hydroxycotinine were significantly associated with 103 and 33 CpGs in maternal and cord sera, respectively (Supplementary Table 1).”. What we meant from this sentence was that nicotine metabolites (cotinine, norcotinine, and 3-hydroxycotinine) in maternal serum were associated with 103 CpGs and the same metabolites in cord serum with 33 CpGs.
To resolve the confusion for readers, we rephrased it to (line 357):
“However, nicotine metabolites (cotinine, norcotinine, and 3-hydroxycotinine) in maternal and cord sera were significantly associated with 103 and 33 CpGs, respectively (Supplementary Table 1).”
Line 373-74 is redundant as it says that "effects of prenatal nicotine exposure on offspring DNAm are exerted through changes in DNAm".
Thank you for the comment. We removed this sentence as the next one addresses the same issue.
In line 377-91, is there literature that shows how methylation changes on the reported genes in utero influence the long term health outcomes of the offspring? As is, not much is said about MYO1G with regards to fetal development and the long term health outcomes. This does not need to be based on maternal smoking.
We are eager analyzing health consequences in particular for allergy and respiratory diseases, however, adding information on health outcomes would go beyond the scope of this paper. We have to present these findings in an additional manuscript.
Also, since the role of methylation on gene transcription regulation is dependent on whether it occurs in the promoter region or gene body, has there been data indicating how smoking affect the direction of gene expression, especially with regard to these 3 genes?
Following your comment, we added information on the location of the CpGs on the genes associated with CpGs to the table 6.
We did not find any clear pattern with regard to the location of CpGs on the gene. Interestingly, the most stable biomarker of the AHRR gene (cg05575921) was not located in the promoter, but the body of the gene.
Also, since lifestyle and environmental factors are known to influence DNA methylation, especially during the fetal development stage, were factors such as nutrition and medications accounted for in the design?
Fetal development and DNAm pattern are influenced by maternal lifestyle or environmental factors such as maternal nutrition or medication use. However, in the presented analysis, we had to focus on one major factor, namely smoking. Other factors such as diet and medication need to be addressed and are part of other manuscripts. In this manuscript we included socioeconomic status, maternal body mass index, and maternal age at delivery as surrogates for differences in the gestational environmental. However, we could exclude SES from the analysis since these were not confounding.
